# Impact of the Duration of Postoperative Antibiotics on the Prognosis of Patients with Infective Endocarditis

**DOI:** 10.3390/antibiotics12010173

**Published:** 2023-01-15

**Authors:** Jinnam Kim, Jung Ho Kim, Hi Jae Lee, Se Ju Lee, Changhyup Kim, Jung Ah Lee, Ki Hyun Lee, Won Kyung Pyo, Jin Young Ahn, Su Jin Jeong, Nam Su Ku, Seung Hyun Lee, Jun Yong Choi, Joon-Sup Yeom

**Affiliations:** 1Division of Infectious Diseases, Department of Internal Medicine, Yonsei University College of Medicine, Seoul 03722, Republic of Korea; 2AIDS Research Institute, Yonsei University College of Medicine, Seoul 03722, Republic of Korea; 3Department of Cardiovascular Surgery, Yonsei University College of Medicine, Seoul 03722, Republic of Korea

**Keywords:** infective endocarditis, postoperative antibiotic therapy, duration, mortality, recurrence, relapse

## Abstract

Appropriate postoperative antibiotic treatment in patients with infective endocarditis (IE) reduces the risks of recurrence and mortality. However, concerns about adverse drug reactions arise due to prolonged antibiotic usage. Therefore, we compared the recurrence and mortality rates according to the duration of postoperative antibiotic therapy in patients with IE. From 2005 to 2017, we retrospectively reviewed 416 patients with IE treated at a tertiary hospital in South Korea; among these, 216 patients who underwent heart valve surgery and received appropriate antibiotics were enrolled. The patients were divided into two groups based on the duration of usage of postoperative antibiotic therapy; the duration of postoperative antibiotic therapy was more than two weeks in 156 patients (72.2%) and two weeks or less in 60 patients (27.8%). The primary endpoint was IE relapse. The secondary endpoints were 1-year IE recurrence, 1-year mortality, and postoperative complication rates. The median age was 53 (interquartile range: 38–62) years. The relapse rate of IE was 0.9% (2/216). There was no statistical difference in relapse (0.0% vs. 1.3%, *p* = 0.379), 1-year recurrence (1.7% vs. 1.3%, *p* = 0.829), or 1-year mortality (10.0% vs. 5.8%, *p* = 0.274) between patients with postoperative antibiotic administration of two weeks or less versus more than two weeks. The duration of postoperative antibiotic therapy did not affect the 1-year mortality rate (log-rank test, *p* = 0.393). In conclusion, there was no statistically significant difference in recurrence, mortality, or postoperative complications according to the duration of postoperative antibiotic therapy.

## 1. Introduction

Infective endocarditis (IE) remains a major medical problem because of its association with high morbidity and mortality; the incidence thereof has increased despite improvements in diagnostic and therapeutic strategies [1,2]. Although surgical intervention plays an important role in reducing mortality, antibiotic therapy is also well established as an important treatment modality for patients with IE [3,4]. Appropriate antibiotic treatment, such as correct dose, duration, and agent selection, can reduce the risk of embolism associated with metastatic infection and poor prognosis in patients with IE [5,6,7]. In addition, appropriate antibiotic treatment is an important factor associated with recurrence and overall mortality in patients with IE [3,8]. However, clinicians tend to prescribe prolonged intravenous antibiotic therapy to patients with IE, because large amounts of bacteria are buried deep in cardiac vegetation, hindering antibacterial access and host immune response [9,10].

Prolonged antibiotic treatment raises concerns about adverse drug reactions and Clostridioides difficile infection (CDI) [11,12,13]. Prolonged parenteral antibiotic treatment dramatically increases the incidence of adverse events such as neutropenia, rash, fever, and eosinophilia [12]. Antibiotics also increase the risk of CDI by disrupting the gut microbiome [13]. Furthermore, prolonged postoperative antibiotic treatment does not have a significant positive effect on recurrence or mortality [14,15]. Rather, short-course postoperative antimicrobial therapy did not result in a significant difference in mortality, relapse, or re-infection in several selected IE cases (even in patients at high risk of complications) [16]. However, unlike total antibiotic duration, no clear guidelines address the duration of postoperative antibiotic therapy, other than to state that a new antibiotic treatment cycle should commence if the valve culture is positive [3].

Therefore, we compared the differences in recurrence, mortality, and major clinical outcomes according to the duration of postoperative antibiotic therapy (≤ or >2 weeks) in patients with IE.

## 2. Results

### 2.1. Patients’ Characteristics

Among 416 patients with IE, 216 patients who underwent heart valve surgery and finished antibiotic therapy were enrolled (Figure 1). The duration of postoperative antibiotic therapy was more than two weeks in 72.2% of the patients (156/216) and two weeks or less in 27.8% (60/216) of the patients.

The median age of the patients was 53 (interquartile range [IQR], 38–62) years, and 67.1% of the patients were male (Table 1). Multiple-valve involvement was observed in 19.9% of IE patients. A previous history of IE was reported in eight patients (3.7%), and a prosthetic valve was reported in 25 patients (11.6%). The median Charlson Comorbidity Index was 1 (IQR, 0–3), and the median European System for Cardiac Operative Risk Evaluation (EuroSCORE) value was 2.04 (IQR, 1.53–2.83). Streptococcus species (42.1%), Staphylococcus species (14.4%), Enterococcus species (6.9%), and Haemophilus, Aggregatibacter, Cardiobacterium, Eikenella, and Kingella [HACEK] (0.5%) were isolated, whereas negative blood culture results were reported in 32.4% of the patients (70/216). Ampicillin/Sulbactam was used in 64 patients (29.6%), penicillin was used in 67 patients (31.0%), and vancomycin was used in 106 patients (49.1%). Two or more antibiotics were used together in 187 patients (86.6%).

There was no statistical difference in age, sex, nosocomial infection, previous IE history, previous history of valve, the affected valves, comorbidities, microorganisms, antibiotics usage, or vegetation size based on the two weeks of postoperative antibiotic therapy. However, the duration of overall antibiotic treatment was longer in those who received more than two weeks of postoperative therapy (28 days vs. 34 days, *p* < 0.001).

### 2.2. Outcomes According to the Duration of Postoperative Antibiotic Therapy

The median follow-up duration was 73 (IQR, 46–109) months. The IE relapse rate was 0.9% (2/216) (Table 2). There was no statistical difference in relapse (0.0% vs. 1.3%, *p* = 0.379), 1-year recurrence (1.7% vs. 1.3%, *p* = 0.829), 1-year reoperation (1.7% vs. 1.9%, *p* = 0.901), or 1-year mortality (10.0% vs. 5.8%, *p* = 0.274) according to postoperative antibiotics duration. The trend in 1-year mortality between groups was determined by the Kaplan–Meier curve and log-rank test (*p* = 0.393) (Figure 2).

No statistically significant difference in postoperative complications was identified between the two postoperative antibiotics duration groups of more than two weeks and two weeks or less; the results are as follows: new-onset heart failure (HF) (16.7% vs. 8.3%, *p* = 0.075), new conduction abnormality (3.2% vs. 10.3%, *p* = 0.099), and new paravalvular complications (10.0% vs. 16.0%, *p* = 0.258) (Table 2).

### 2.3. Comparison of the Postoperative Antibiotic Therapy Duration According to the 1-Year Composite Outcome in Patients with Infective Endocarditis Who Underwent Valve Surgery

The median duration of postoperative antibiotic treatment was 21 (IQR: 14–28) days in IE patients who underwent valve surgery and lacked the 1-year composite outcome, but 20.5 (IQR: 5–34) days in IE patients with the 1-year composite outcome (*p* = 0.386). There was no statistically significant difference between the two groups (Figure 3). The variability of postoperative antibiotic duration is presented in Appendix A.

### 2.4. Univariable and Multivariable Logistic Regression Analysis of the 1-Year Composite Outcome in Patients with Infective Endocarditis

Upon univariable analysis, Staphylococcus species were related to an increase in the 1-year composite outcome (*p* = 0.026) (Table 3). Upon multivariable analysis, Staphylococcus species was also associated with an increasing 1-year composite outcome (odds ration [OR] 3.683, 95% confidence interval [CI] 1.341-10.114, *p* = 0.011). Postoperative antibiotics duration did not show a statistically significant association with the 1-year composite outcome.

## 3. Discussion

In our study, the duration of postoperative antibiotic therapy did not show a statistically significant difference in relapse, 1-year recurrence, 1-year reoperation, 1-year mortality, or postoperative complications in IE patients who underwent valve surgery. The postoperative antibiotic duration was not statistically significant different according to the 1-year composite outcome in IE patients who underwent valve surgery.

Both surgical intervention and appropriate antibiotic therapy are important for patients with IE. In patients with IE who receive appropriate antibiotic treatment, the risk of embolic complication is reduced to less than half [5]. Embolism leads to ischemic stroke, mycotic aneurysms, and other metastatic infarctions, and is associated with IE recurrence and poor prognosis [3,6,7]. Inadequate antibiotic treatment due to the incorrect selection of dose, duration, or agent is a crucial factor associated with an increased rate of recurrence [3]. Thuny F. et al. reported that recurrence is an independent predictor of increased mortality in patients with IE [8]. Consequently, appropriate antibiotic treatment can reduce the risk of recurrence and mortality in patients with IE.

The total antibiotic duration is well specified for each species in the guidelines [3]. However, regarding the duration of postoperative antibiotics, it is recommended to start a new antibiotic treatment cycle when the valve culture is positive, but other opinions are not clear. Gisler et al. reported that the most prominent effect of preoperative antibiotic treatment on valve culture results was observed within the first few days, and did not significantly reduce the risk of positive valve culture beyond 21 days [17]. Morris et al. reported that prolonged postoperative antibiotic treatment (4–6 weeks) was not associated with the recurrence rate of IE, and two weeks seemed sufficient in the absence of metastatic infection [15]. Rao et al. also reported that a postoperative antibiotic course less than two weeks was not associated with recurrence and survival in patients with IE [14]. Thus, the need for prolonged postoperative antibiotics in IE patients who undergo valve surgery has been questioned. In our study, there was no significant difference in the types and duration of antibiotic treatment between the two groups of patients with more than and less than two weeks’ usage of postoperative antibiotic therapy, respectively. However, prolonged postoperative antibiotic treatment beyond two weeks did not show a significant effect on recurrence or mortality in patients with IE who underwent appropriate surgical intervention. A logistic regression analysis, after adjusting for various confounders, showed that prolonged postoperative antibiotic treatment did not show a significant effect on 1-year mortality. Furthermore, the duration of postoperative antibiotics did not affect the 1-year composite outcome in patients with infective endocarditis who underwent valve surgery.

There are concerns about prolonged use of antibiotics due to adverse drug reactions and CDI. Vancomycin has been associated with ototoxicity, nephrotoxicity, neutropenia, and thrombocytopenia, especially in adults who are receiving prolonged therapy [11]. Prolonged parenteral β-lactam therapy for more than two weeks in IE patients dramatically increases the incidence of delayed adverse events such as neutropenia, rash, fever, and eosinophilia [12]. Antibiotics also disrupt the normal gut microbiome and increase the risk of CDI [13]. Considering that an increase in adverse drug reactions and prolonged postoperative antibiotic treatment do not have significant effects on recurrence rate and mortality, an appropriate evaluation according to the duration of postoperative antibiotics therapy is necessary.

Inadequate antibiotic treatment, positive valve culture, prosthetic valve IE, persistent metastatic foci of infection, and empirical antimicrobial therapy for blood culture-negative IE are reported as factors that increase the rate of relapse [3,18]. In our study, we encountered a relapse in only two patients, one of whom had IE with multiple valve involvement and septic shock, and the other had a metastatic infection in a cardiac device. A shorter course of postoperative antibiotic therapy could be considered when employing a multidisciplinary approach, that is, when most vegetation is removed and neither a metastatic infection nor a positive valve culture is noted [3,14,15].

Staphylococcus species constitute an important causative pathogen in native and prosthetic valve endocarditis. Staphylococcus aureus is a common cause of acute and destructive IE and is an independent risk factor for relapse, in-hospital mortality, and 1-year mortality [19,20,21]. Coagulase negative staphylococci causes more protracted valve infections and increases the risk of in-hospital death and relapse in patients with IE [22,23,24]. In our study, the presence of Staphylococcus species was also a significant risk factor associated with the 1-year composite outcome in patients with infective endocarditis who underwent valve surgery.

Our study had certain limitations. First, due to the retrospective nature of the study, the baseline characteristics of the two groups classified by the postoperative antibiotic duration were not completely homogeneous. Second, as this was a single-center study, the sample size was small. Third, since cases that are inoperable or do not meet surgical indication were excluded, the severity or bacterial etiologies could be different. Fourth, patients with recurrent endocarditis who died before diagnosis, or cases where this was not followed up may be missed. However, most variables that could affect the outcomes did not have a statistically significant difference. In addition, a logistic regression analysis was performed to exclude the influence of confounders that could affect the clinical outcomes.

## 4. Materials and Methods

### 4.1. Patient Selection

From November 2005 to August 2017, we retrospectively reviewed 416 patients with IE at a single tertiary hospital in South Korea. The inclusion criteria were as follows: patients aged 18 years or older who were diagnosed with IE according to the modified Duke criteria, and who underwent heart valve surgery thereafter. Patients who did not require surgical treatment, who died before the end of treatment, who evidenced positive valve culture, or who completed the course of antibiotic therapy prior to surgery were excluded. Patients who received a prolonged course of antibiotics, such as for tuberculosis or a fungal infection, were also excluded. Only the first IE episode was counted in one patient, and subsequent events were described as recurrence.

This study was approved by the institutional review boards (IRBs) of Yonsei University College of Medicine (IRB no. 4-2018-0248). Due to the retrospective nature of the study, informed consent was waived. This study complied with the Good Clinical Practice guidelines and the Declaration of Helsinki.

### 4.2. Variables Definition

IE was defined according to the modified Duke criteria, and included both “definite IE” and “possible IE” [25]. Antibiotics were selected according to the European Society of Cardiology (ESC) guidelines and Infectious Diseases Society of America guidelines [3,26]. Appropriate antibiotic treatment was considered when at least one of the compounds was effective against the causative microorganism and when administered intravenously; specifically, if the identified strain was not susceptible to the empirically initiated antibiotics, the duration of antibiotic use was not counted until the date of change according to the susceptibility [17,27]. Decisions on surgical intervention were recommended by the multidisciplinary team, according to the guidelines [3,26]. A previous history of valves included prosthetic valves, previous valve surgery, cardiac devices, and other structural causes. Recurrence included both relapse and reinfection; “relapse” referred to repeated episodes of IE by the same microorganism, while “reinfection” referred to repeated episodes of IE by a different microorganism or later events [28]. Reoperation was defined as additional surgery on the same heart valve not only for the IE recurrence, but also for postoperative valve complications [14]. The 1-year composite outcome, including 1-year recurrence, 1-year reoperation, and 1-year mortality, was evaluated instead of evaluating each outcome individually, because there were few recurrence cases. The Charlson comorbidity index is a weighted index to estimate the risk of death from a comorbid disease at hospital admission [29]. The EuroSCORE was used for surgical risk stratification according to the patient’s demographic characteristics, cardiovascular and non-cardiovascular risk factors, and procedural variables [30]. The mortality data were obtained from the Ministry of the Interior and Safety of South Korea.

### 4.3. Primary and Secondary Endpoints

The primary endpoint was IE relapse. The secondary endpoints were 1-year IE recurrence, 1-year mortality, valve-related reoperation, and postoperative complications, such as new-onset HF, conduction abnormality, and paravalvular complications.

### 4.4. Statistical Analysis

Patients were divided into two groups according to the duration of postoperative antibiotic therapy: more than and less than two weeks. A comparison between the groups was analyzed using the Chi-square and Fisher’s exact test for categorical variables and the Mann–Whitney U test for continuous variables. A *p*-value of <0.05 was considered statistically significant. Kaplan–Meier (K-M) curves showed the probability of the survival of patients from the time of admission (due to IE) to either death, or the date of last known follow-up (18 September 2019) within one year. The log-rank test was used to explore any differences in the probability of death between the two groups. Scatter plots were used to display the distribution pattern of postoperative antibiotic duration according to the 1-year composite outcome. A logistic regression analysis was performed to further adjust for confounding factors and to analyze ORs and 95% CIs in terms of 1-year mortality. The variables for the multivariable analysis were selected based on clinically significant risk factors with *p* < 0.05 in the univariable analysis. The statistical analysis was performed using IBM SPSS Statistics for Windows, Version 26 software (IBM Corp. Armonk, NY, USA).

## 5. Conclusions

In conclusion, there was no statistically significant difference in the recurrence or mortality rates according to the duration of postoperative antibiotic therapy. If surgery and recommended antibiotic duration are properly performed, a shorter duration of postoperative antibiotics could be considered. Therefore, further larger prospective studies are warranted to assess the effect of postoperative antibiotic treatment duration on patients with IE.

## Figures and Tables

**Figure 1 antibiotics-12-00173-f001:**
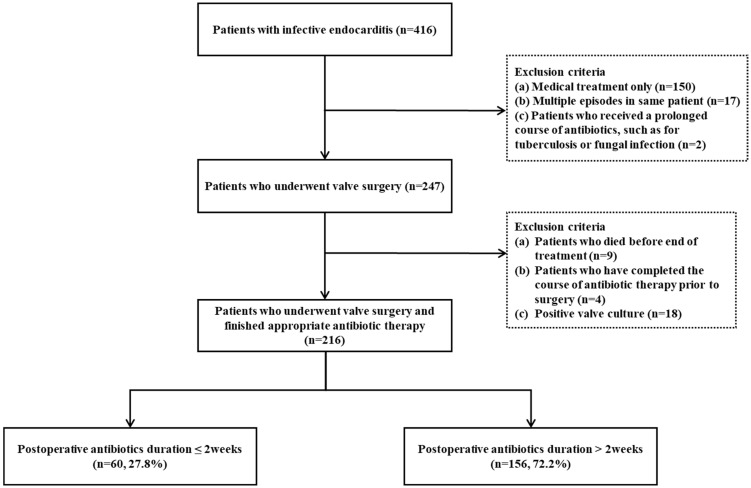
Flow chart of study patients with infective endocarditis.

**Figure 2 antibiotics-12-00173-f002:**
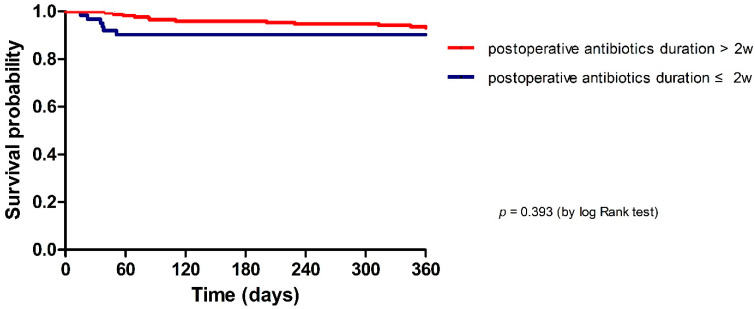
Kaplan–Meier curves of 1-year mortality in patients with infective endocarditis according to the duration of postoperative antibiotic usage.

**Figure 3 antibiotics-12-00173-f003:**
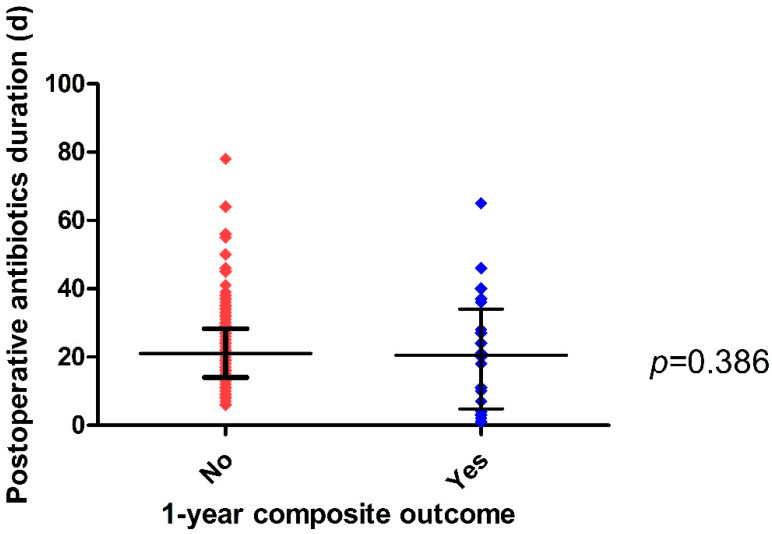
Scatter plot of postoperative antibiotic treatment duration according to the 1-year composite outcome in patients with infective endocarditis who underwent valve surgery.

**Table 1 antibiotics-12-00173-t001:** Baseline characteristics of patients with infective endocarditis who underwent valve surgery.

Variable	Total (n = 216)	Postoperative Antibiotics Duration		*p* Value
		≤2wks (n = 60, 27.8%)	>2wks (n = 156, 72.2%)	
Age (years)	53 (38–62)	53 (35–66)	52 (38–62)	0.762
<60	151 (69.9%)	41 (68.3%)	110 (70.5%)	
60–80	62 (28.7%)	18 (30.0%)	44 (28.2%)	
>80	3 (1.4%)	1 (1.7%)	2 (1.3%)	
Male sex	145 (67.1%)	39 (65.0%)	106 (67.9%)	0.679
Nosocomial infection	20 (9.3%)	3 (5.0%)	17 (10.9%)	0.180
Previous infective endocarditis	8 (3.7%)	2 (3.3%)	6 (3.8%)	0.999
Previous history of valves	85 (39.4%)	21 (35.0%)	64 (41.0%)	0.417
Prosthetic valve	25 (11.6%)	5 (8.3%)	20 (12.8%)	0.356
Previous valve surgery	32 (14.8%)	8 (13.3%)	24 (15.4%)	0.704
Cardiac devices	3 (1.4%)	0 (0.0%)	3 (1.9%)	0.562
Affected valve				
Aortic valve	104 (48.1%)	25 (41.7%)	79 (50.6%)	0.237
Mitral valve	144 (66.7%)	46 (76.7%)	98 (62.8%)	0.053
Tricuspid valve	9 (4.2%)	3 (5.0%)	6 (3.8%)	0.711
Pulmonary valve	5 (2.3%)	1 (1.7%)	4 (2.6%)	0.999
Multiple valves	43 (19.9%)	14 (23.3%)	29 (18.6%)	0.434
Other comorbidities				
Diabetes mellitus	29 (13.4%)	7 (11.7%)	22 (14.1%)	0.638
Chronic heart failure	9 (4.2%)	2 (3.3%)	7 (4.5%)	0.999
End stage renal disease	2 (0.9%)	0 (0.0%)	2 (1.3%)	0.999
Liver disease	9 (4.2%)	0 (0.0%)	9 (5.8%)	0.065
Solid cancer	12 (5.6%)	3 (5.0%)	9 (5.8%)	0.999
Hematologic malignancy	2 (0.9%)	1 (1.7%)	1 (0.6%)	0.479
Connective tissue disease	6 (2.8%)	2 (3.3%)	4 (2.6%)	0.671
Immunosuppressive therapy	5 (2.3%)	1 (1.7%)	4 (2.6%)	0.999
Central venous access	5 (2.3%)	0 (0.0%)	5 (3.2%)	0.325
Charlson Comorbidity Index	1 (0–3)	1 (0–3)	1 (0–3)	0.548
EuroSCORE value	2.04 (1.53–2.83)	1.76 (1.53–2.82)	2.06 (1.53–2.83)	0.556
Clinical signs and symptoms (initial)				
Fever (≥38 °C)	153 (70.8%)	42 (70.0%)	111 (71.2%)	0.867
Left ventricular dysfunction (EF < 50%)	79 (36.6%)	19 (31.7%)	60 (38.5%)	0.353
Sepsis (including septic shock)	141 (65.3%)	39 (65.0%)	102 (65.4%)	0.958
Skin lesions	3 (1.4%)	1 (1.7%)	2 (1.3%)	0.999
Embolic complications				
CNS embolic complications	64 (29.6%)	19 (31.7%)	45 (28.8%)	0.684
Renal failure	18 (8.3%)	5 (8.3%)	13 (8.3%)	0.999
PAOD	2 (0.9%)	0 (0.0%)	2 (1.3%)	0.379
Other systemic emboli	18 (8.3%)	5 (8.3%)	13 (8.3%)	0.999
Microbiology				
Coagulase negative staphylococci	17 (7.9%)	6 (10.0%)	11 (7.1%)	0.573
Staphylococcus aureus	14 (6.5%)	4 (6.7%)	10 (6.4%)	0.999
MSSA	10 (4.6%)	3 (5.0%)	7 (4.5%)	0.999
MRSA	4 (1.9%)	1 (1.7%)	3 (1.9%)	0.999
Enterococcus species	15 (6.9%)	3 (5.0%)	12 (7.7%)	0.765
Streptococcus species	91 (42.1%)	26 (43.3%)	65 (41.7%)	0.824
HACEK	1 (0.5%)	1 (1.7%)	0 (0.0%)	0.278
Gram negative bacilli (except HACEK)	3 (1.5%)	0 (0.0%)	3 (1.9%)	0.562
Others	10 (4.6%)	2 (3.3%)	8 (5.1%)	0.730
Culture-negative	70 (32.4%)	19 (31.7%)	51 (32.7%)	0.885
Antibiotics				
Ampicillin/Sulbactam	64 (29.6%)	14 (23.3%)	50 (32.1%)	0.209
Penicillin	67 (31.0%)	20 (33.3%)	47 (30.1%)	0.648
Other β-lactams ^a^	123 (56.9%)	35 (58.3%)	88 (56.4%)	0.798
Vancomycin	106 (49.1%)	25 (41.7%)	81 (51.9%)	0.177
Aminoglycoside	162 (75.0%)	48 (80.0%)	114 (73.1%)	0.293
Quinolone	1 (0.5%)	0 (0.0%)	1 (0.6%)	0.999
Use of more than one antibiotic	187 (86.6%)	52 (86.7%)	135 (86.5%)	0.980
Duration of antibiotic treatment				
Duration of ampicillin/sulbactam (days)	26 (11–39)	17 (11–26)	27 (13–40)	0.096
Duration of penicillin (days)	29 (25–35)	28 (18–35)	29 (26–35)	0.361
Duration of other β-lactams (days)	27 (14–40)	17 (14–28)	17 (5–36)	0.059
Duration of vancomycin (days)	24 (8–34)	16 (12–28)	12 (5–35)	0.920
Duration of aminoglycoside (days)	21 (14–29)	26 (10–31)	21 (14–32)	0.849
Patients with vegetation (initial)	202 (93.5%)	56 (93.3%)	146 (93.6%)	0.999
Median maximal vegetation size (cm)	1.1 (0.7–1.6)	1.1 (0.7–1.8)	1.1 (0.7–1.6)	0.805

Continuous variables are described as median and interquartile range (IQR), and discrete variables are described as numbers (%). ^a^ Other β-lactams comprised nafcillin, ceftriaxone, cefazolin, or piperacillin/tazobactam. EuroSCORE: European system for cardiac operative risk evaluation; EF: Ejection Fraction; CNS: Central Nerve System; PAOD: Peripheral arterial occlusive disease; MSSA: Methicillin-susceptible Staphylococcus aureus; MRSA: Methicillin-resistant Staphylococcus aureus; HACEK: Haemophilus, Aggregatibacter, Cardiobacterium, Eikenella, and Kingella.

**Table 2 antibiotics-12-00173-t002:** Postoperative outcomes in patients with infective endocarditis who underwent valve surgery.

Postoperative Outcomes	Total (n = 216)	Postoperative Antibiotics Duration		*p* Value
		≤2wks (n = 60, 27.8%)	>2wks (n = 156, 72.2%)	
1-year Recurrence ^a^	3 (1.4%)	1 (1.7%)	2 (1.3%)	0.829
Relapse ^b^	2 (0.9%)	0 (0.0%)	2 (1.3%)	0.379
Reinfection ^c^	1 (0.5%)	1 (1.7%)	0 (0.0%)	0.107
1-year Reoperation ^d^	4 (1.9%)	1 (1.7%)	3 (1.9%)	0.901
1-year Mortality	15 (6.9%)	6 (10.0%)	9 (5.8%)	0.274
1-year Composite outcome ^e^	20 (9.3%)	8 (13.3%)	12 (7.7%)	0.201
New-onset heart failure	23 (10.6%)	10 (16.7%)	13 (8.3%)	0.075
New conduction abnormality	18 (8.3%)	2 (3.3%)	16 (10.3%)	0.099
New paravalvular complications	31 (14.4%)	6 (10.0%)	25 (16.0%)	0.258

^a^ Recurrence: defined as both relapse and reinfection. ^b^ Relapse: defined as at least one repeat episode of IE caused by the same microorganism within 6 months. ^c^ Reinfection: defined as subsequent IE caused by a different microorganism. ^d^ Reoperation: defined as the need for additional surgery on the same heart valve, not only to treat recurrent IE but also valve complications. ^e^ 1-year composite outcome comprised the 1-year recurrence, reoperation, and mortality rates.

**Table 3 antibiotics-12-00173-t003:** Univariable and multivariable logistic regression analysis of the 1-year composite outcome in patients with infective endocarditis.

Characteristics	N	Univariable Analysis			Multivariable Analysis		
		HR	95% CI	*p*-Value	HR	95% CI	*p*-Value
Sex							
Male	145	1					
Female	71	1.387	0.495–3.887	0.534			
Previous infective endocarditis	8	0.700	0.064–7.646	0.770			
Prosthetic valve	25	1.912	0.519–7.035	0.330			
Multiple valve involvement	43	2.253	0.740–6.864	0.153			
Charlson comorbidity index		1.110	0.940–1.311	0.220			
Microbiology							
Staphylococcus species ^a^	32	3.648	1.166–11.413	0.026	3.683	1.341–10.114	0.011
Culture-negative ^b^	70	1.184	0.357–3.931	0.783			
Postoperative antibiotics duration (days)		0.987	0.947–1.029	0.536			

^a^ Staphylococcus species comprised the coagulase negative staphylococci and Staphylococcus aureus. ^b^ Culture-negative: defined as negative blood culture results.

## Data Availability

The data presented in this study are available upon reasonable request from the corresponding author.

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
