# Peer review of "Impact of the Duration of Postoperative Antibiotics on the Prognosis of Patients with Infective Endocarditis"

_antibiotics, 2023, doi:10.3390/antibiotics12010173_

Round 1
Reviewer 1 Report
I read the manuscript with interest. The study was written very carefully. The sample size is not apparently large, but the main advantage of the study is long FU. The aim of the study is clearly defined - to compare the effectiveness of shorter and longer antibiotic therapy in patients with IE undergoing surgical treatment. While in lead related infective endocarditis the guidelines quite precisely specify the recommended duration of antibiotic therapy after lead extraction, the duration of antibiotic therapy after cardiac surgery is not clearly specified in the recommendations (due to the lack of research). The study has a chance to cite the recommendations for the treatment of IE in the next edition.
I have no critical comments
Author Response
As stated in your comments, "unlike total antibiotic duration, no clear guidelines address the duration of postoperative antibiotic therapy, other than to state that a new antibiotic treatment cycle should commence if the valve culture is positive" was the starting point of this study. Thank you for your comments on this study.
Reviewer 2 Report
This study addresses a very important issue in the treatment of infective endocarditis, namely the length of the post-operative antibiotic treatment. The study is of some interest but the methodology is very problematic in many respects. Below I give my main concerns and other comments.
Main concerns
1. The study cohort is composed of all patients that underwent surgery. I think that the relevant cohort should consist of the patients that underwent surgery AND finished antibiotic treatment (thus excluding those who died before end of treatment). The patients with positive valve cultures could also be excluded since the relevant population to study are the ones with negative valve cultures. The authors excluded those who received shorter treatment than prescribed in guidelines and this is not acceptable. Those should be included.
2. I believe that the primary outcome should not be death. Death can be due to many reasons. What we want to avoid by post-operative antibiotic treatment in IE is relapse in bacteremia or endocarditis. This would be the relevant outcome measure, much more relevant than death. Moreover, death within the observation period is very likely not to be related to infective endocarditis and is not a well-chosen outcome. Death within a year after end of treatment is much more relevant and could be a secondary outcome measure.
3. In table 1 the dicotomized cohort is shown. The cohort is divided in two uneven parts and this is highly rpoblematic. This limits the possibilities to demonstrate potential differences. It is very unlikely that 2 weeks is some kind of magical limit. It would be a much more precise and relevant way to describe the cohort to give the median treatment time (with IQR) for the dichotomized variables and put continuous variables in categories and give treatment times for the respective categories. Differences could be investigated using Mann-Whitney u tests for dichotomous variables such as sex and with Kruskall-Wallis test for variables such as age divided in three categories. Table 1 would thus give median treatment time for eg men and women respectively and then investigate if there is a difference with MWU-test. For age, the median time for those aged over 80, 60-80 and below 60 would be given and difference would be investigated with KW-test…
4. Figure 2, table 2 and 3 should be deleted.
5. The main outcomes should be given as a scatterplot where the postoperative treatment time is given for those with the main outcome (namely recurrence of bacteremia or endocarditis within one year (five patients- though it is not really possible to understand if these had infection with the same pathogen (relevant) or another pathogen (which will render them a non-outcome)) and those that survived for one year without outcome. Comparison should be made with MWU-test. As secondary outcome death within one year after end of postoperative treatment can be tried and postoperative treatment time for those who died and those who lived can be compared.
6. The time between end of treatment and relapse should be given and details of those with relapse should be presented. Main demography, type of valve and pathogen and treatment time as a minimum. This should be possible as these patients were few.
7. In table three post-operative outcomes are mixed with preoperative features like embolism and heart failure. This is completely misleading and must be changed.
8. Discuss how the very different bacterial etiologies in the cohort compared to most contemporary cohorts (almost no S. aureus and a large proportion of culture negative IE) affect generalizability of the results.
Other comments
1. Title: use “patients with infective endocarditis” not “infective endocarditis patients”.
2. There are claims for example in abstract that postoperative antibiotic treatment is important to hinder embolisms. This is a strange claim. Antibiotics are given to avoid recurrence. I never saw postoperative embolisms. Of course you can suffer stroke during surgery but risk for embolism after surgery should be very low.
3. Abstract “increase in the incidence of resistant strains”. The use of “incidence” and “strains” is incorrect and the potential selection for antibiotic resistant bacteria within the patient is of very limited importance in this respect. Antibiotic resistance is of course a huge problem but the added pressure of the few patients operated for endocarditis contributes in very limited way to this. Other side-effects such as clostridoides infection are however highly relevant. Change.
4. Bacterial species names should be given in italics and with a space between the abbreviated genus name and species name. Check throughout.
5. Introduction “limited host immune response indicate a trend”. How does this indicate a trend? I do not understand. Rephrase
6. “The total antibiotic duration is well specified for each strain in the guideline”. The authors mean “species” not “strain”. Change
7. Lines 166-179, delete, not related to this study.
8. The claim “Prosthetic valve is a major risk factor for IE that should require antibiotic prophylaxis 184 prior to dental procedures [3].” is completely unrelated to the present study. Delete!
9. 199-206 not relevant and in parts not correct. Delete
10. “Antibiotics were selected according to the European Society of Cardiology (ESC) guidelines [3].” This claim is bizarre for a cohort which was gathered from 2005 and onwards since guidelines were published in 2015.
11. Line 235 “was not susceptible to the identified strain”. It is not the bacterium that has effect on the antibiotic but the other way around. Rephrase.
12. Line 239-241 discuss reinfection and recurrence. I see no result where this difference is given…
Round 2
Reviewer 2 Report
The manuscript has been substantially improved. Some things, however, remain to be revised.
1. In abstract it is not clear to which groups the different proportions refer. Please clarify.
2. On different places in the manuscript different claims about the two main groups are given. The authors need to be extremely clear if the patients that received exactly 14 days of postoperative therapy belong to the “short” or “long” therapy group. Make appropriate changes.
3. The variability of length in the main exposure variable (length of postoperative antibiotic treatment) should be accounted for. Could be a supplementary figure. Scatterplot would be good.
4. Describe the two patients with the primary outcome in terms of microbiology, valve involvement, total and postoperative antibiotic treatment. Time and type of relapse.. Was it even the same CoNS? Same pattern of resistance? Same species?
5. Table 2. In this table outcome measures are combined with preoperative features of the patients, or at least I believe that this is the case. New onset heart failure, new conduction abnormality, paravalvular complication and embolic complication certainly did not occur after conclusion of post-operative antibiotic therapy!?? Therefore these features should be moved to table 1.
6. Heading of section 2.3. Change “scatter plot” to “Comparison”
7. Table 3, what is the reference to Staphylococcus and blood culture negative?
8. Line 196-197, this is driven by mortality almost exclusively. Please rephrase.
9. Is there a risk that a recurrent endocarditis can be missed? The patient died without diagnosis? Referred to another hospital at recurrence? This should be acknowledged as a limitation.
